# UV-exposure, endogenous DNA damage, and DNA replication errors shape the spectra of genome changes in human skin

Natalie Saini[1¤], Camille K. Giacobone[1], Leszek J. Klimczak[2], Brian N. Papas[2], Adam B. Burkholder[2], Jian-Liang Li[2], David C. Fargo[2], Re Bai[3], Kevin Gerrish[3], Cynthia L. Innes[4], Shepherd H. Schurman[4], Dmitry A. Gordenin[1]*

1 Genome Integrity and Structural Biology Laboratory, National Institute of Environmental Health Sciences, US National Institutes of Health, Research Triangle Park, North Carolina, United States of America, 2 Integrative Bioinformatics Support Group, National Institute of Environmental Health Sciences, US National Institutes of Health, Research Triangle Park, North Carolina, United States of America, 3 Molecular Genomics Core Laboratory, National Institute of Environmental Health Sciences, US National Institutes of Health, Research Triangle Park, North Carolina, United States of America, 4 Clinical Research Unit, National Institute of Environmental Health Sciences, US National Institutes of Health, Research Triangle Park, North Carolina, United States of America

¤ Current address: Department of Biochemistry and Molecular Biology, Medical University of South Carolina, Charleston, South Carolina, United States of America

* gordenin@niehs.nih.gov

**Data Availability Statement:** All BAM and MAF files are available under controlled access from the dbGaP database (phs001182.v2.p1 https://www.

## Abstract

Human skin is continuously exposed to environmental DNA damage leading to the accumulation of somatic mutations over the lifetime of an individual. Mutagenesis in human skin cells can be also caused by endogenous DNA damage and by DNA replication errors. The contributions of these processes to the somatic mutation load in the skin of healthy humans has so far not been accurately assessed because the low numbers of mutations from current sequencing methodologies preclude the distinction between sequencing errors and true somatic genome changes. In this work, we sequenced genomes of single cell-derived clonal lineages obtained from primary skin cells of a large cohort of healthy individuals across a wide range of ages. We report here the range of mutation load and a comprehensive view of the various somatic genome changes that accumulate in skin cells. We demonstrate that UV-induced base substitutions, insertions and deletions are prominent even in sun-shielded skin. In addition, we detect accumulation of mutations due to spontaneous deamination of methylated cytosines as well as insertions and deletions characteristic of DNA replication errors in these cells. The endogenously induced somatic mutations and indels also demonstrate a linear increase with age, while UV-induced mutation load is age-independent. Finally, we show that DNA replication stalling at common fragile sites are potent sources of gross chromosomal rearrangements in human cells. Thus, somatic mutations in skin of healthy individuals reflect the interplay of environmental and endogenous factors in facilitating genome instability and carcinogenesis.

ncbi.nlm.nih.gov/projects/gap/cgi-bin/study.cgi?study_id=phs001182.v2.p1). All other data including the underlying numerical data for all of graphs and summary statistics are in Supplementary Tables. The R-code for analysis of the trinucleotide-specific mutation signatures can be accessed via https://github.com/NIEHS/P-MACD".

**Funding:** This work was supported by the US National Institute of Health Intramural Research Program Project Z1AES103266 to D.A.G. The funders had no role in study design, data collection and analysis, decision to publish, or preparation of the manuscript.

**Competing interests:** The authors have declared that no competing interests exist.

## Author summary

Skin forms the first barrier against a variety of environmental toxins and DNA damaging agents. Additionally, DNA of skin cells suffer from endogenous damage and errors during replication. Altogether, these lesions cause a variety of genome changes resulting in disease including cancer. However, the accurate measurement of the range and complete spectrum of genome changes in healthy skin was missing due to technical or biological limitations of prior studies. We present here accurate measurements of the various types of somatic genome changes that we found in skin fibroblasts and melanocytes from 21 donors ranging in ages from 25 to 79 years, which allowed to distinguish age related from age independent changes. Our cohort contains both White and African American donors, allowing an estimation of the impacts of skin color on mutagenesis. As a result, we revealed the complete spectrum and determined the range of somatic genome changes and their etiologies in healthy human skin fibroblasts and melanocytes and highlighted molecular mechanisms underlying these changes. Therefore, our study introduces a base line for defining disease levels of genome instability in skin.

## Introduction

Cells within the human body encounter a vast variety of DNA damaging agents throughout an individual's lifetime. By some estimates, cells may receive 70,000 DNA lesions per day [1,2]. Erroneous repair or lack of repair of these lesions would lead to a variety of genome changes including somatic single base substitutions, insertions and deletions, rearrangements and copy number changes. Large-scale sequencing studies of single cells, clonally expanded single cells and bulk cells from healthy humans have demonstrated that healthy human tissues are genetically mosaic with thousands of somatic mutations [3–11]. Analysis of such accumulated somatic genome changes have enabled elucidation of the sources of the mutation-initiating lesions as well as the various DNA repair pathways that may be involved in error-prone repair of DNA damage in human cancers [12–15]. Since at least half of the somatic genome changes seen in cancers originate in healthy pre-cancerous cells [16], it is imperative to establish the sources of DNA damage and their impacts on genome stability in healthy cancer-free tissues.

Skin is the largest tissue in the human body and forms the first line of defense against environmental toxins and DNA damaging agents, with ultraviolet (UV) radiation being the most potent environmental mutagen in skin cells. In fact, melanoma genomes have the highest burdens of mutations with UV-induced mutation signatures predominating amongst the mutation signatures identified in this cancer type [12,13]. The pathogenic impact of UV-radiation in generating genome instability is multifaceted. UV-induced DNA lesions are a source of replicative polymerase stalling [17,18] and require translesion synthesis (TLS) over cyclobutane pyrimidine dimers (CPD) and pyrimidine 6–4 pyrimidone (6-4PP) [19–27]. Error-prone TLS over UV-induced lesions leads to C➜T changes in the yCn motif (y is any pyrimidine, n is any nucleotide, mutated base is capitalized). Cytosines within a CPD may also be deaminated to uracils and upon copying by the canonical DNA polymerases or by the TLS polymerase, Pol η, would be fixed as yCn➜yTn changes or to CC➜TT changes in the next round of replication [19,21]. Error-prone TLS across thymine CPDs can also lead to T➜C changes [19,21,28,29] preferring nTt➜nCt motif [8]. Altogether, these base-substitution motifs derived from experimental data constitute a significant part of mutation signature SBS7b extracted by non-negative matrix factorization analysis from mutation catalogs of thousands of whole-genome sequenced human cancers [13]. In the absence of TLS across UV-induced lesion it would not result in base substitutions but can lead to impediment of replication fork progression. Restart

of a stalled replication fork can result in the formation of single-stranded gaps in the sister DNA molecules and later convert to double strand breaks (DSBs) [30,31]. Inaccurate repair of such a DSB via homologous recombination (HR) or non-homologous end joining (NHEJ) can lead to a structural changes, copy number variation or generate a small insertion or a deletion.

In agreement with UV radiation being the major source of DNA damage in skin cells, various studies have demonstrated that C➔T changes in the yCn context is the most prevalent base substitution in skin fibroblasts, melanocytes, and keratinocytes. In addition, human skin cells also carry CC➔TT changes and T➔C in the nTt motifs [7,8,32,33]. Moreover, we previously demonstrated that fibroblasts obtained from sun-exposed body sites carry a higher mutation burden along with a higher contribution of a UV-mutation signature than fibroblasts obtained from sun-shielded sites [8]. Our findings were also supported in the study by Tang *et*. *al*. wherein they demonstrated higher mutation burden in melanocytes from sun-exposed body sites than sun-shielded body sites via either whole exome sequencing or targeted sequencing of 509 cancer-associated genes in single melanocytes [33]. In summary, numerous studies have established and verified the prominent mutagenic effects of the bypass of UV-induced lesions by translesion polymerases generating a characteristic base substitution signature in skin cells. However, the broad spectrum of somatic genome instability, including consequences of UV-induced DSBs in cells of healthy human skin have neither been established nor characterized by mutation signature analysis.

In addition to environmental DNA damage, cells may also accumulate somatic genome changes due to endogenous DNA damage or errors during DNA replication in the form of base substitutions, small insertions or deletions (indels), and gross chromosomal rearrangements. Somatic mutations in skin cells have been measured either by deep sequencing of bulk tissue [32] or whole-genome sequencing of single cell-derived induced pluripotent stem cells [5] or single cell-derived clonal lineages [7,8]. However, due to either small sample sizes or difficulties in accurately identifying somatic indels and chromosomal rearrangements using induced pluripotent stem cells or bulk cells, none of these studies have been able to adequately characterize the different sources of DNA damage and their mutagenic outcomes in skin cells from healthy donors [34].

Here, we present an integrated analysis of the various types of somatic genome changes that are found in skin fibroblasts and melanocytes from a total of 21 donors ranging in ages from 25 to 79 years. Unlike previous studies, our cohort contains both White and African American donors, allowing a better estimation of the impacts of skin color on mutagenesis in skin cells. Our work provides the normal range of the burden and types of somatic genome instability in human skin cells. We show here that in skin cells, endogenous DNA damage in the form of spontaneously deaminated cytosines at CpG motifs, oxidative DNA damage, as well as DNA replication errors, are a substantial source of somatic mutagenesis. Additionally, UV-induced DNA damage is prevalent even in sun-shielded skin cells and manifests as single base substitutions arising from DNA synthesis over lesions by TLS and by deletions of five or more nucleotides arising from end-joining repair of UV-induced DSBs. Our analysis also highlights the differences in the outcomes of UV-induced DSBs and DSBs induced by endogenous DNA damage in cancer-free skin cells. Overall, we provide a comprehensive analysis of the various UV-induced and endogenous genome de-stabilizing processes that operate in healthy skin cells.

## Results

### Study design

Based on our prior study [8], we performed whole-genome sequencing of hip skin cells as it would allow detection of versatile mutational processes, because it is mostly sun-shielded. UV-

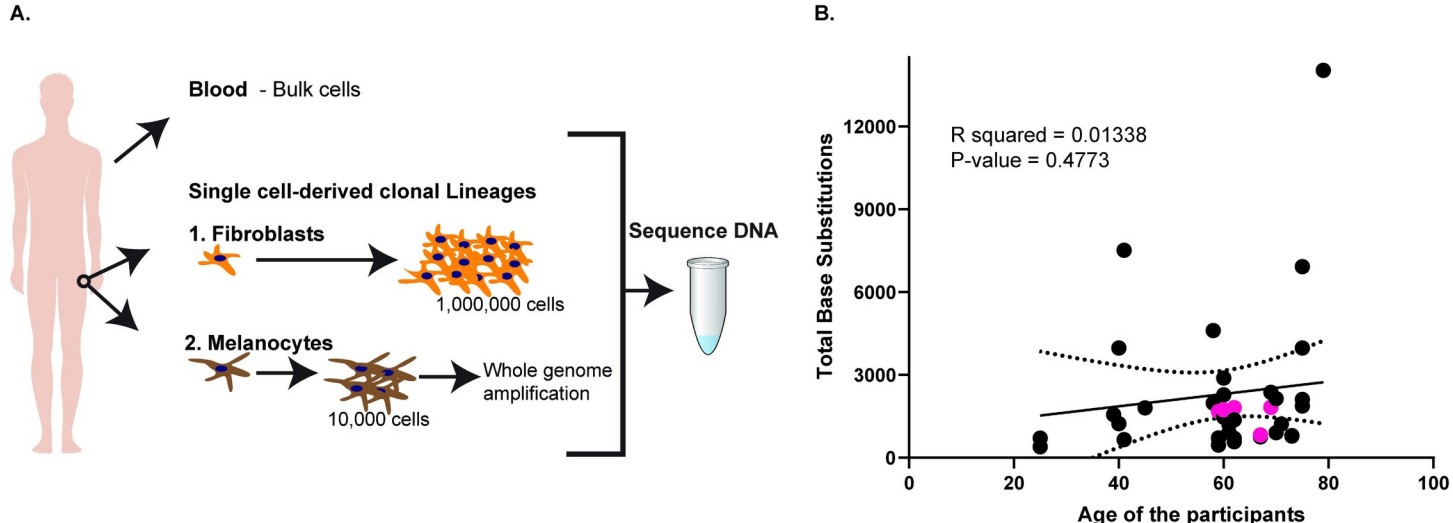

**Fig 1. Schematics and total base substitutions identified per clonal lineage in this study.** (A) Schematics of the study design. From each donor, we obtained blood for whole-genome sequencing. In addition, we obtained skin biopsies from the hips of the donors from which fibroblasts and melanocyte clonal lineages were obtained. Fibroblasts were grown up to a million cells and their DNA was directly used for whole-genome sequencing, while melanocytes grew up to 10,000 cells and the DNA was whole-genome amplified and thereafter sequenced. (B) The total base substitutions in each clonal lineage versus the age of the donors. The pink filled circles denote melanocyte clones. The x-axis denotes the ages of the donors, while the y-axis denotes the number of base substitutions. The solid black line is the linear regression line for the samples, while the dotted black curves are the 95% confidence intervals. The source data for this figure is in S1 and S2 Tables.

induced mutagenesis has lower contribution into overall mutation spectrum in this tissue which allows better detection of other types of mutagenesis. In this study, we analyzed somatic genome changes in single cell lineages from 34 fibroblasts obtained from skin biopsies taken from hips of a total of 21 donors, ages 25 to 79. Our dataset includes the hip fibroblasts from two donors sequenced previously [8]. In addition, we sequenced five genomes of clonal single melanocyte-derived lineages. Single skin fibroblasts were propagated in culture up to approximately 1,000,000 cells which provided sufficient high-quality genomic DNA for whole-genome sequencing and follow-up validation. The clonal single-melanocyte lineages were cultured in media up to 10,000 cells. DNA from these cells was whole-genome amplified and sequenced. In addition, we were able to grow one melanocyte clone up to 1,000,000 cells and performed whole-genome sequencing on this sample without whole-genome amplification. From each donor, we also sequenced whole blood DNA (Fig 1A).

The median sequencing depth for the samples was 78X with a minimum average coverage per site of 50X (S1 Table). The genome-wide changes detected in the clones were compared to blood samples from the same donors and only the variants unique to the clones were denoted as somatic changes in the clones. Stringent filtering criteria were applied to exclude changes that could have occurred during limited propagation of the clone. For this purpose, only base substitutions as well as indels calls within 45% and 55% (heterozygous alleles) or above 90% (homozygous alleles) allele frequencies were considered clonal and somatic in the initiating melanocyte or fibroblast cell. All other calls that did not conform to these allele frequencies were considered sub-clonal and were removed from the analysis as these most likely represented culture-induced artifacts. We also analyzed the allele frequencies of all somatic base substitutions in the clonal lineages. All fibroblast clonal lineages demonstrated a peak of mutation calls at the 45% to 55% allele frequencies, indicating that these samples were clonal (S1 Fig and S2 Table). We did not see such a peak in the whole-genome amplified melanocyte clones which could reflect uneven genome amplification and localized genome duplications during the whole-genome amplification step. Nonetheless, only analyzing heterozygous mutation

calls within the 45% and 55% allele frequencies and homozygous mutation calls at >90% allele frequencies allows us to estimate the minimum number of somatic mutations in the founder cells. Mutations that accrue during culture and/or polymerase errors during whole genome amplification are expected to not be clonal and have allele frequencies <45%. For structural changes, clonal calls with variant junction reads representing at least 30% of the total junction reads, and no reads representing the variants in the blood genomes were identified as clonal somatic rearrangements present in the initiating fibroblast. Multiple samples sequenced from the same donors allowed intra-individual comparisons of somatic genome instability in humans.

## UV-induced base substitutions and C➔T changes due to spontaneous cytosine deamination are prevalent in skin cells

We detected 402 to 14029 base substitutions in each clonal lineage sequenced and the mutations did not increase with the age of the donors (Fig 1B). Analysis of the mutation spectrum revealed that the predominant mutation in many samples was C➔T base substitution (S2 Fig and S2 Table). The number and types of base substitutions in melanocyte clones were similar to those seen in the fibroblasts (Figs 1B and S2 and S2–S4 Tables). To identify the predominant mutation signatures in our samples, we determined the cosine similarities of the 96 tri-nucleotide motif mutation profiles in our samples versus all the published mutation signatures derived from analysis of thousands of mutation catalogs from human tumors ([13], and https://cancer.sanger.ac.uk/cosmic/signatures). This allowed us to agnostically determine the mutation signatures previously identified in cancers that were also overrepresented in our samples. We saw that the SBS7b signature was overrepresented in many samples. These signatures comprise of C➔T changes at cC or tC motifs (S2 and S3 Figs). In addition, SBS1 was only weakly represented in our samples. We also found mutation signatures SBS2 and SBS11 present in the samples which also carry a strong SBS7b mutation signature. These are likely due to the overlap between the SBS2 and SBS11 mutation signatures with UV-induced mutations (S3 Table). We also used non-negative matrix factorization (NMF)-based deconvolution of mutation signatures as a parallel approach to agnostically determine the predominant signatures in our samples for single base substitutions and dinucleotide substitutions. We were able to detect SBS1, SBS5, and prominent UV mutation signatures (SBS7b and DBS1) in our cohort (S3 Table, S2 and S3 Figs). In 28 samples we also detected either SBS4 or SBS18 which are indicative of oxidative damage in the cells leading to G➔T (C➔A) changes (S2 and S3 Figs and S3 Table).

We then sought to determine if the most prominent components of mutational signatures identified above were statistically enriched in our samples. For this purpose, we used our previously described knowledge-based trinucleotide-motif-centered pipeline [8,14,15,35]. This pipeline calculates enrichment with mutations within pre-defined trinucleotide motifs. It also calculates the sample-specific p-values for enrichments and minimum estimate of mutation load assigned to a motif-specific mutagenic process after stringent statistical filtering. nCg➔nTg changes likely arise upon spontaneous deamination of methylated cytosines [15]. These mutations constitute the major component of SBS1 in COSMIC [12,13]. SBS1-associated mutation load has been shown to increase with age in cancers [36] and in healthy individuals [7,10,37,38]. Analysis of the nCg➔nTg changes in our donors demonstrated that this mutation type is statistically enriched in all the samples and was also found to linearly increase with the ages of the participants with an average increase of 0.4 mutations per year (Fig 2 and S4 Table). We also detected statistically significant enrichment with UV-associated C➔T changes in the tC or cC context (yCn➔yTn, major component of COSMIC SBS7b) in many of

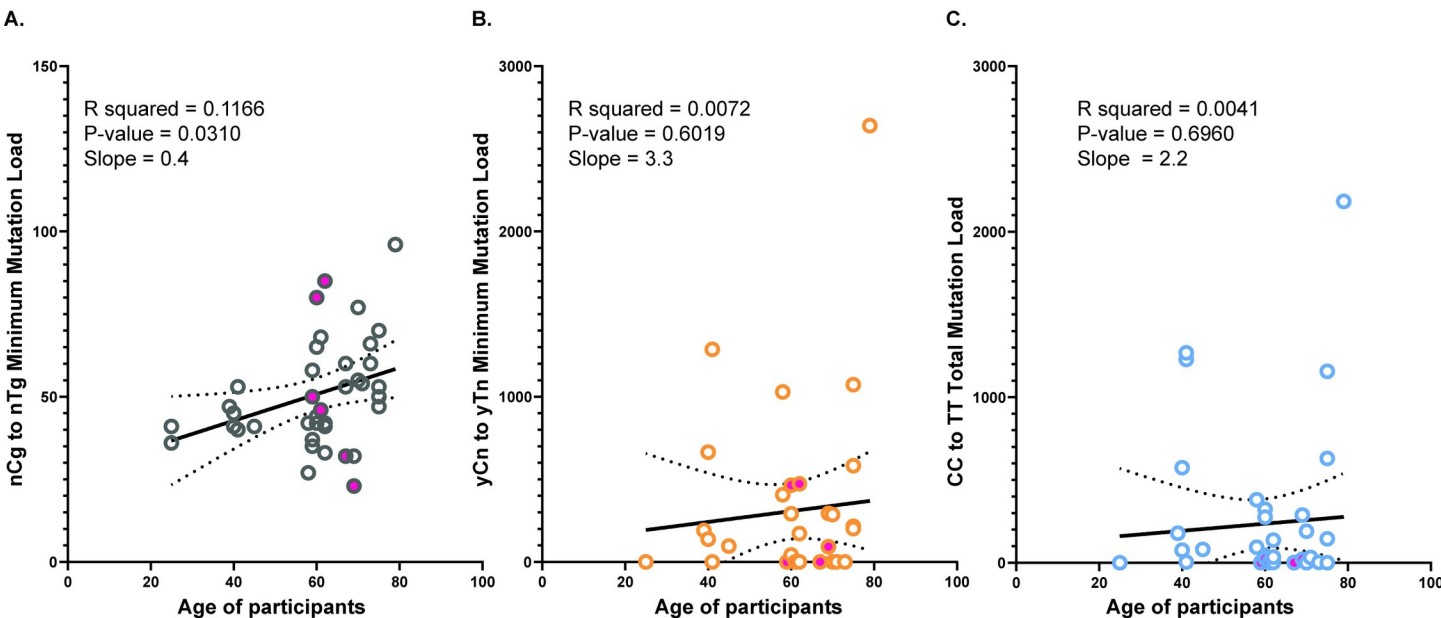

**Fig 2. Analysis of the motif-specific mutation signatures in the genomes of skin cells.** The minimum mutation load for the (A) nCg➔nTg mutation signature, (B) yCn➔yTn mutation signature and (C) the total mutation load for the CC➔TT dinucleotide changes are plotted against the ages of the participants. The solid pink circles denote the mutation load in melanocytes. The black solid line is the linear regression, and the dotted curves are the 95% confidence intervals for each dataset. The source data for this figure is S4 Table.

the sequenced samples as well as a prominent presence of CC➔TT changes (Fig 2 and S4 Table). The estimates of yCn➔yTn minimum mutation load correlate with direct counts of the less frequent CC➔TT and nTt➔nCt changes, which have previously been shown to be associated with UV-induced DNA damage in human cells [8,13] (S4 Fig). Altogether these three types of changes indicate the contribution of UV-induced changes in mutation load accumulated in skin. Interestingly, the UV-induced mutations did not correlate with the ages of the participants (Fig 2).

SBS1 mutations identified by SigProfilerExtractor (NMF-based deconvolution) and the minimum number of nCg➔nTg mutations analyzed by the knowledge-based pipeline correlate with each other. We saw a similar correlation between the mutations attributed to SBS7b by SigProfilerExtractor and the minimum number of yCn➔yTn mutations identified by the knowledge-based pipeline (S5 Fig). These data indicate that both methodologies perform similarly in the evaluation of mutation signatures.

## Bulk exome sequencing reveals the presence of cancer drivers in the samples at less than 10% allele frequencies

Whole-exome sequencing up to 100X to 150X was also performed for bulk samples from which 14 fibroblast clones and four melanocyte clones, respectively (S1 Table). Analysis of single nucleotide variants (SNVs) in the bulk samples and comparisons with the clonal lineages derived from the bulk cells revealed the presence of overlapping SNVs in bulk and the corresponding clones. Interestingly, we did not see any overlapping SNVs between bulk samples and clonal lineages that were not derived from the same bulk sample, even if they were coming from the same donor (S5 Table). This observation validates our mutation calling pipeline and provides support for the presence of the mutations detected in the clonal lineages in the original skin biopsies.

The somatic mutations identified in the bulk samples were predominantly at or less than 10% allele frequencies (S5 Table and S6 Fig). This observation was also found to hold true for the allele frequencies of the overlapping SNVs in bulk samples and their corresponding clonal lineages (S7 Fig). The low allele frequency in the population demonstrates the large amount of heterogeneity in the dermal and the epidermal tissue.

We also annotated all the SNVs in all whole-genome sequenced clones and whole-exome sequenced bulk tissues for functional effects. All non-synonymous SNVs, stop gains, start or stop loss SNVs in the clones and bulk tissues were further analyzed using the Cancer Genome Interpreter [39] to determine if these were potentially cancer drivers. Of the 672 SNVs in the clones that had potentially functional impacts, 32 changes were in tumor driver genes and 13 changes were annotated as tumor drivers. One sample, DAG_H95, was found to have 3 tumor driver mutations, however the donor does not have any history of cancer (S2 Table). We also detected 3190 SNVs in the bulk tissues that could alter protein sequence, of which 390 were within tumor driver genes, and 62 of the mutations were annotated as driver mutations. Interestingly, these driver mutations were also present between 2 to 10% allele frequencies in the samples (S7 Fig). Overall, the results suggest that normal sun-shielded human skin carries a substantial proportion of cancer driver mutations, albeit at low allele frequencies.

## Single base indels in homonucleotide repeats and deletions larger than 5 bases are ubiquitous in skin cells

We detected from 7 to 71 indels in the donors (Fig 3A and S6 Table). The insertions ranged from 1 base to 40 bases and deletions ranged from 1 to 171 bases (Fig 3B and S6 Table). The total number of indels per sample do not appear to increase statistically with the ages of the donors (Fig 3A). NMF-based deconvolution analysis of indel signatures or measuring the cosine similarities of indel patterns with the indel signatures currently annotated in cancers [13] demonstrated that two indel types were prevalent in our samples. The first was single base insertions or deletions in homopolymeric stretches associated in our samples with ID1, ID2 and ID7 (S8 Fig and S7 and S8 Tables). Since many samples had very low numbers of indels, it is possible that mathematical deconvolution of indel signatures may carry errors. Therefore, instead of the number of mutations within each signature, we used the total number of single base insertions or deletions in homopolymeric stretches for further downstream analyses. These types of indels were also found to increase linearly with the ages (0.22 mutations per year) of the donors consistent with the idea that they were associated with polymerase slippage at the homopolymeric repeats [40,41] during ongoing DNA replication in fibroblasts over the donors' lifetime (Fig 3C). The second class of indels were deletions spanning five nucleotides or more, many of which have microhomology of one or more bases at the deletion junction (S8 Fig and S8 Table). Based on cosine similarities, these indels were highly similar to ID8 (S8 Table), the indel signature associated with double-strand break repair via non-homologous end joining [13]. Consistently, NMF-based deconvolution of indel signatures applied to our samples identified these deletions of five or more bases as part of novel signature which is a composite signature made up of ID8-like indels as well as indels in homopolymeric repeats (Signature A in S8 Fig and S8 Table). Such deletions spanning five or more nucleotides were identified in almost all samples and did not demonstrate a statistically significant increase with the ages of the donors. We also did not see any differences between the indel load or signatures in melanocytes versus the fibroblasts indicating that the processes yielding indels in both cell types are likely the same (Fig 3C).

The number of deletions spanning five or more nucleotides were found to also correlate in our samples with the UV-associated trinucleotide-centered yCn to yTn mutation signature.

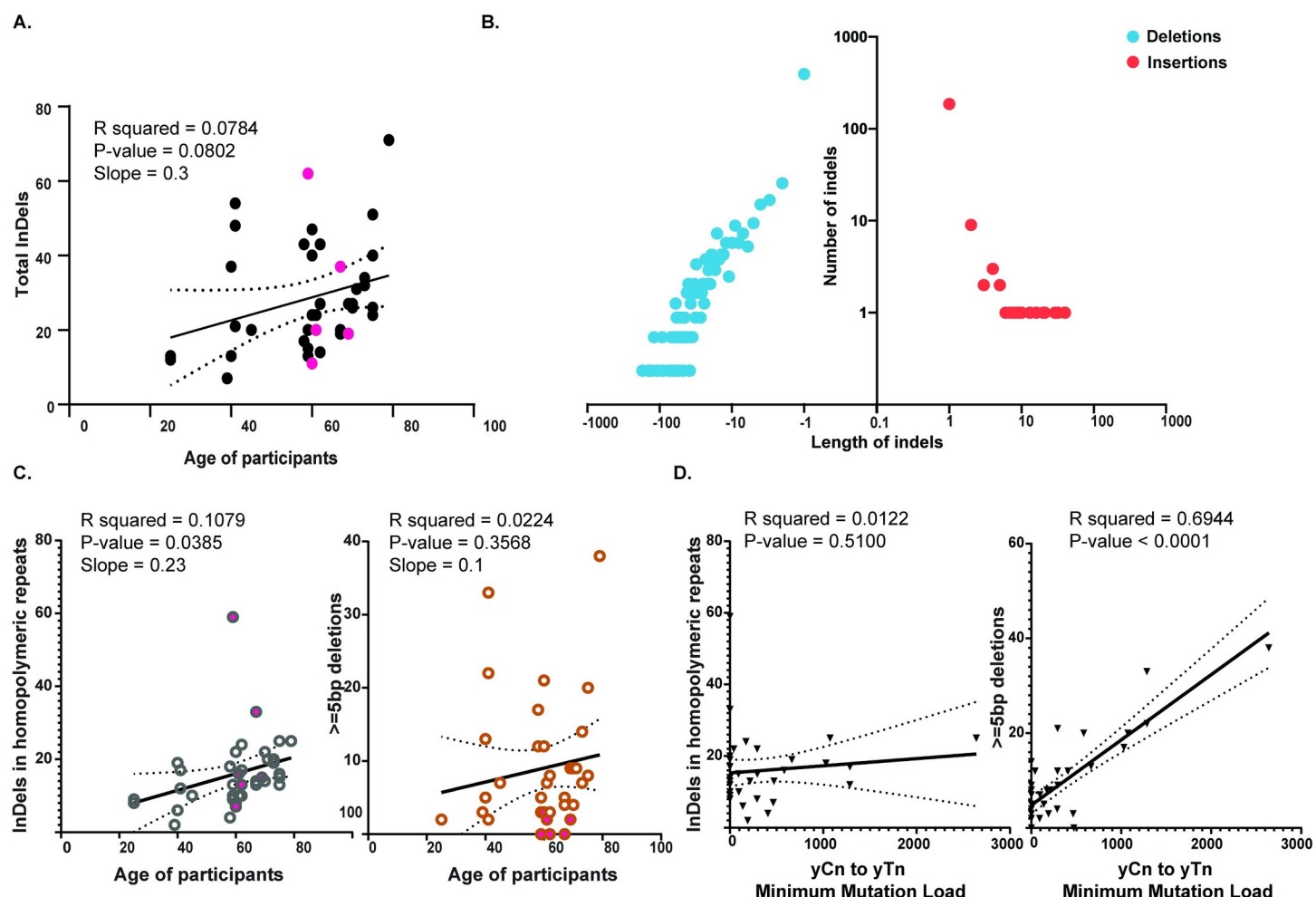

**Fig 3. Analyses of indels in skin cells.** (A) The total number of indels identified in each sample plotted against the ages of the donors. The black dots denote fibroblast clones, while the pink dots denote the melanocyte clones. (B) The distribution of the lengths of the insertions and deletions detected in the clonal lineages. The source data for panels A and B is in S6 Table. (C) The number of insertions and deletions in homopolymeric repeats and the deletions spanning 5 bases or more plotted against the ages of the donors. The open circles denote fibroblast clones, while the filled in circles denote melanocyte clones. The source data for this figure are in S7 Table. (D) The number of insertions and deletions in homopolymeric repeats and the deletions spanning five bases or more plotted against the yCn➜yTn minimum mutation load in skin cells. The source data for this panel are in S4 and S6 Tables. In the graphs, the black solid line is the linear regression of the data, and the dotted black curves are the 95% confidence intervals.

We did not see a positive correlation between the UV-associated mutation signatures and single base indels in homopolymeric repeats (Fig 3D). These data indicate that unlike indels at homopolymeric repeats, UV-induced DNA double strand breaks are the underlying etiology for deletions of five or more bases in human skin cells.

## The majority of the insertions in skin cells are templated

The predominant insertions detected in our clonal lineages were templated single-base insertions (i.e. copied from the neighboring bases). Of the 186 single base insertions, 163 of the insertions were copied from the adjacent base (S9 Table). Such insertions most likely represent polymerase slippage events within homopolymeric runs of bases or erroneous Okazaki fragment maturation [40,41] and constitute the ID1, ID2 and ID3 indel signatures as mentioned above [13].

We also detected 28 instances of insertions larger than two bases in length. 18 of these larger insertions were a duplication of the neighboring residues. Three of these templated insertions carried small mismatches likely due to errors during copying of the neighboring residues (S9 Table). Such templated insertions along with deletions spanning five bases or more have been shown to be characteristic of non-homologous or microhomology-mediated end-joining of double-strand breaks [13,42,43]. As such, it is likely that repair of UV-induced double strand breaks also leads to insertions of two bases and more. However, the low numbers of such events do not allow statistical verification of this hypothesis.

## UV-induced mutation load varies by race and is not impacted by the sex of the donors

Our cohort included five African American or Black donors and 16 White donors, thus allowing us to also determine if the accumulation of somatic genome changes is different between the two races. The total base substitutions in samples from the White donors (median 1824), were higher than in the skin fibroblasts and melanocytes obtained from African American or Black donors (median 715, p-value = 0.00002193, calculated by two-tailed Mann Whitney test). We reasoned that this lower mutation load in the African American donors might reflect the protective effect of melanin in skin. Consistently, we did see a prominent presence of UV-associated yCn➡yTn changes in skin cells from White donors (median for White donors = 209). However, we did not detect statistically significant enrichment with this mutation type in skin cells from Black donors (minimum estimate of mutation load = 0, Fig 4A and S10 Table). The number of nCg➡nTg mutations, which are not associated with UV-lesions did not vary across the two categories of donors (median for White donors = 48, median for African American donors = 40). In addition, although we did not see any difference in the total number of indels or the number of indels in homopolymeric repeats in skin cells obtained from donors of either race, we found increased numbers of deletions spanning five bases or more in skin cells obtained from White donors (median = 9) as compared to the skin cells obtained from African American or Black donors (median = 2, P-value = 0.0002781, calculated by two tailed Mann Whitney test) (Fig 4B and S10 Table). In order to avoid skewing of the data due to differences in sequencing methodologies used for melanocytes and fibroblasts in our samples, we also calculated P-values for each of the cohorts using a two tailed Mann-Whitney test after excluding the data from the melanocytes. Even in this data set, we were clearly able to detect an increase in UV-induced mutations in White donors as compared to African American or Black donors (S10 Table). Overall, our data can be explained by melanin in skin providing strong protection against UV-associated somatic mutations in the form of both UV-signature base substitutions as well as deletions of five bases or more.

Our cohort also consists of eight men and 13 women. Analysis of mutation load based on sex did not demonstrate any differences between the skin cells obtained from the men or the women (S9 Fig and S10 Table).

## Structural variant hotspots colocalize with common fragile sites

Structural variants were only analyzed for the fibroblast clonal lineages and the single melanocyte clonally grown lineage for which we were able to obtain sufficient cells for WGS without whole-genome amplification, since the genome amplification process can result in many false rearrangement calls. There were 120 structural variants in the 35 sequenced clonal lineages (from 1 to 14 in each isolate) (Fig 5A and S11 Table). The structural variants included deletions, duplications, inversions ranging in size from 225 bp to 39 Mbp, as well as translocations. No age-dependent increase in structural variants was evident.

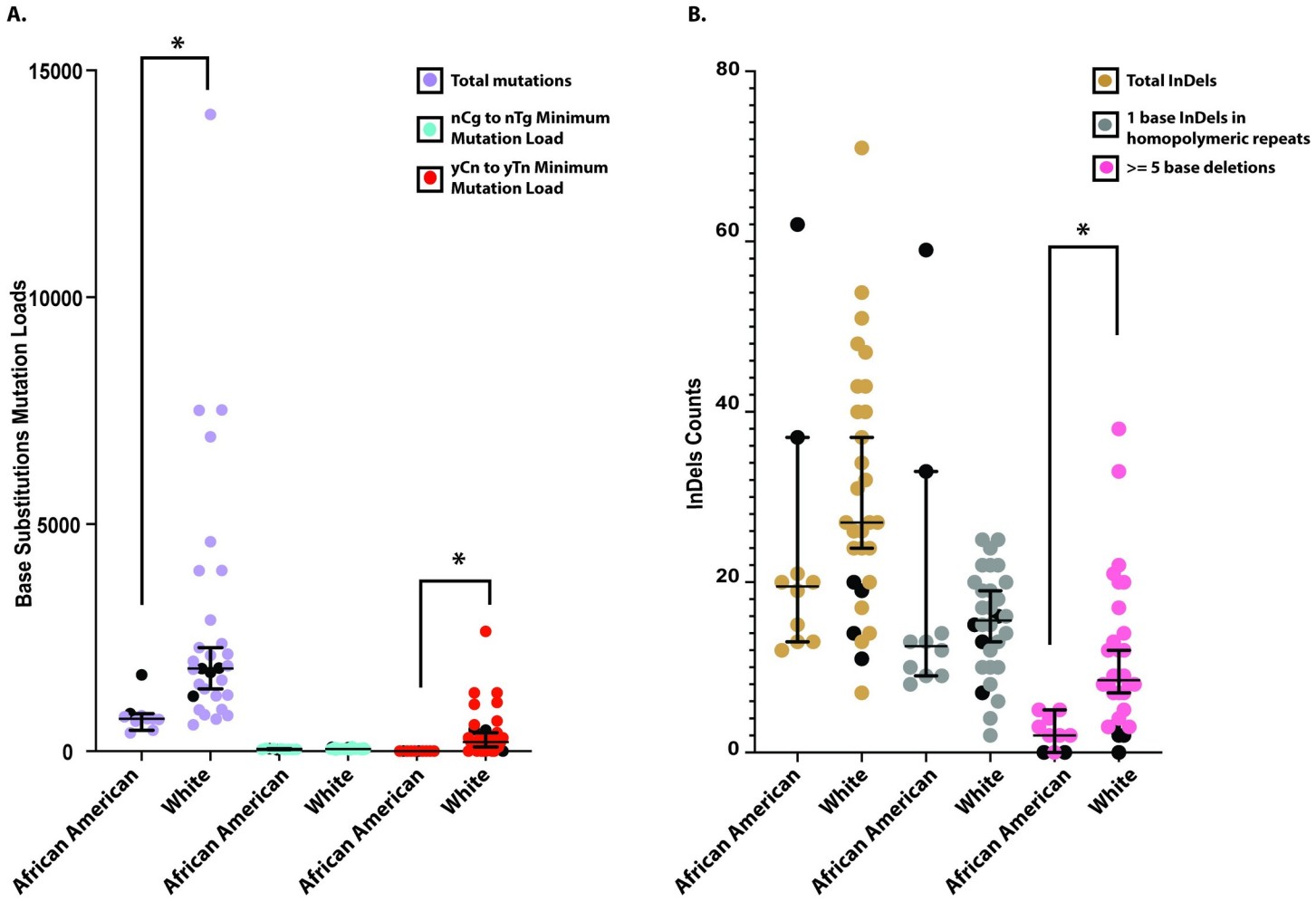

**Fig 4. Base substitutions and indels in African American and White donors.** (A) The total number of base substitutions, the nCg➔nTg minimum mutation load and the yCn➔yTn minimum mutation load in African American and White donors. Melanocytes are depicted as filled black circles. (B) The total number of indels, single nucleotide indels in homopolymeric repeats and deletions spanning five bases or more in the African American and White donors. Melanocytes are depicted as filled black circles. A two-sided Mann-Whitney U-test was performed to compare the mutation load across the two cohorts. * denotes a Bonferroni corrected P-value < 0.05. The source data for this figure is in S10 Table.

We identified genomic regions that are hotspots for chromosomal breakage and structural variation. Two or more rearrangements were denoted as part of a hotspot if they were less than or equal to 1Mbp apart and were present in different samples. Of the 120 structural variants identified, 55 rearrangements were within hotspots. Previously, we showed that structural variants identified in skin fibroblasts of two donors were often in the vicinities of common fragile sites (CFSs) [8]. To determine if the structural variants in this larger data set also often colocalize with CFSs, we identified those deletions, duplications and inversions that intersect common fragile sites within the HumCFS database [44]. We also identified those translocations as colocalizing with fragile sites, whose breakpoints were within 10kb of a CFS. 63 rearrangements were found to colocalize with CFSs. 18 of the rearrangements within CFSs were on chromosome 7, of which 14 rearrangements were within FRA7J, implying that this fragile site is expressed more prominently in fibroblasts than the other fragile sites, leading to higher levels of replication stalling and gross chromosomal breakage. Moreover, the majority of the rearrangements within hotspots also colocalized with CFSs, while the majority of

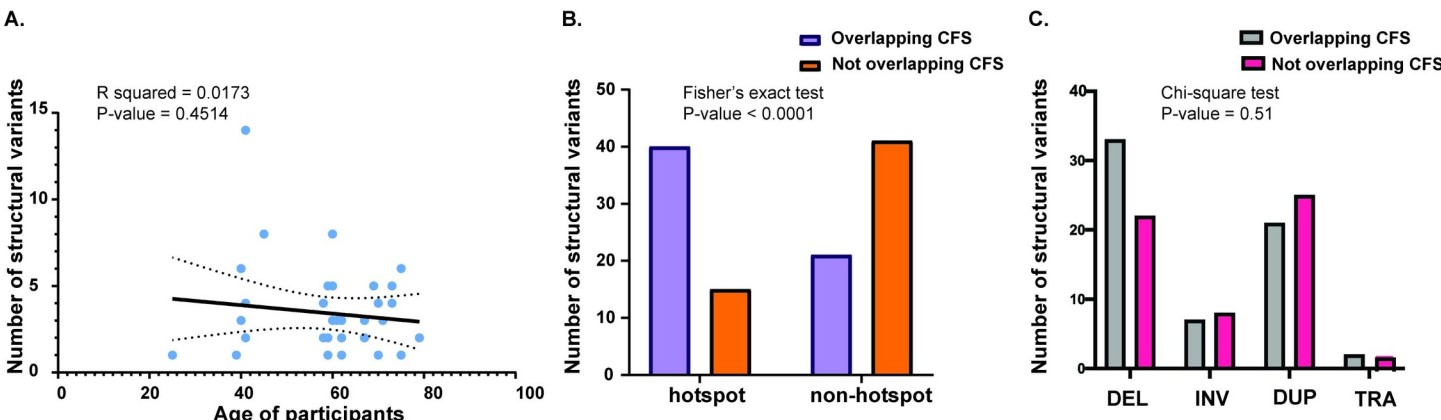

**Fig 5. The structural variants identified in the genomes of skin cells.** (A) The number of structural variants in each donor plotted against the ages of the donors. The black inclined line denotes the linear regression of the data, while the dotted curves denote the 95% confidence intervals. (B) The number of structural variants that were or were not within hotspots and common fragile sites. A Fisher's exact test was performed to determine if structural variants in hotspots were also preferentially present within common fragile sites. (C) The types of structural variants that overlap and do not overlap common fragile sites. A Chi-square test was performed to determine if the structural variant types within common fragile sites were different from those that did not overlap common fragile sites. The source data for this figure are in S11 Table.

rearrangements that were not within hotspots were scattered across the genome (Fig 5B and S11 Table). Interestingly, we did not see any difference in the types of structural variants that overlap and those that did not overlap CFSs (Fig 5C). Moreover, we determined the use of microhomology at the breakpoints to identify a role of microhomology mediated repair of DSBs at CFSs. Of the 120 rearrangements, only 15 rearrangements contained microhomology at the breakpoints (6 overlapping CFSs, and 9 not overlapping CFSs). These regions of microhomology were small and ranged from 2 to 3 bases. A Fisher's exact test demonstrated no significant bias in the use of microhomology between the variants that overlapped CFSs versus those that do not (P-value 0.41) (S11 Table).

Overall, we hypothesize that replication-associated difficulties at CFSs are responsible for the generation of rearrangement hotspots in healthy human cells.

## Discussion

In this study, we revealed and accurately measured load of the major types of somatic genome changes in human skin. We grew single cell-clonal lineages derived from human skin fibroblasts and melanocytes. Whole genome sequencing from these samples allows us to detect somatic genome changes that are present in the original single cells with high accuracy. Moreover, the methodology provides sufficient DNA for orthogonal validation of the changes, allowing us to apply the most stringent criteria for identifying different kinds of genome changes without losing sensitivity.

Our work provides the range of normal somatic genome changes in human skin cells in donors across a wide range of ages and of different races. We demonstrate each skin cell carries from 402 to 14029 base substitutions, 7 to 71 indels and 1 to 14 structural variants per cell. The mutation burden in healthy skin cells was also similar to the median mutation load in cancers [45]. Interestingly, we identified various cancer driver mutations in the clones as well as in the bulk tissue samples, although these driver mutations were present at low allele frequencies in the bulk samples. This observation echoes previous findings where normal tissue often contains cells with driver mutations [11,32,37,38,46,47].

Analysis of mutation signatures in the clonal lineages allowed to differentiate between endogenous DNA damage-induced mutations, replication-associated errors as well as

environmental DNA damage-induced genome changes. C➔T changes at CpG motifs as well as single nucleotide insertions and deletions were found to increase with the ages of the donors and were indicative of endogenous mutational processes and replication errors, respectively. In addition, UV-induced base substitution signatures were prominent in many samples even though they were obtained from sun-shielded skin. UV-induced DNA damage can also lead to the formation of double strand breaks in the genome. We showed here that deletions spanning 5 or more nucleotides with or without microhomologies at the junctions strongly correlated with the UV-induced base substitution signature. Previously, this indel signature (ID8) has been identified in a wide variety of cancers, and likely represents repair of double strand breaks via non-homologous end joining (NHEJ) pathways [13]. As such, we hypothesize that ID8-like indels are characteristic of UV damage in human cells. Since we also detect deletions with limited microhomologies at the junctions, it is possible that in addition to canonical NHEJ, micro-homology mediated end joining (MMEJ) or polymerase theta-mediated end joining (TMEJ) [48,49] may also participate in the repair of UV-induced DSBs in skin cells. In addition to dele-tions, we also detected a few instances of long insertions, often templated from the flanking sequences in our samples. Such locally templated insertions are highly characteristic of TMEJ and are likely formed by the Polθ-dependent synthesis wherein one resected DSB end uses the second resected DSB end for synthesis [42]. Overall, our data indicates that ID8-like indels along with a small number of templated insertion events accumulate in skin cells, due to UV-induced DNA damage and error-prone repair via NHEJ or TMEJ.

Interestingly, although non-UV mutations (nCg➔nTg and indels at homopolymeric repeats) increased with the ages of the participants, we did not see a similar correlation between UV-exposure-induced mutations and ages of the donors. Since we are measuring mutation load in sun-shielded skin cells, as such, even intermittent UV-exposure due to cloth-ing and lifestyle choices of the participants during their lifetimes are likely to lead to the forma-tion of UV-induced DNA damage and impact the lifetime accumulation of UV-induced mutation load in hip-derived cells. Thus, the absence of a correlation between age of the donors and the UV-induced mutation load might be due to differences in overall accumula-tion of UV-exposure across the lifetime of different donors.

DNA double strand breaks can be channeled into repair via two major pathways, HR or NHEJ. One major factor that determines the choice of the repair pathway in cells is the cell cycle stage. Cells in the S or early G2 phases of the cell cycle predominantly repair DSBs via HR, while NHEJ events peak in the G1 or late G2 phases [50,51]. Since HR is mostly error-free, we would not be able to detect HR activity that may have occurred in skin cells. Nonethe-less, the prominent presence of UV-associated NHEJ or TMEJ-generated indels in human skin further indicates that the majority of UV-associated damage and mutagenesis accrues in quies-cent non dividing cells.

We also demonstrated here that UV-associated mutation load is decreased in skin cells from African American donors as compared to White donors. We hypothesize that this effect may be due to the protective effects of melanin on UV-induced DNA damage. In agreement with the decreased mutation load, are the lower rates of skin cancer in African Americans. While skin cancer accounts for up to 35–45% of all cancers in Caucasians [52], it only accounts for 1–2% of the neoplasms in African Americans [53–55]. Moreover, the impact of UV-expo-sure as a risk factor for skin cancers is decreased in African Americans as compared to Cauca-sians [55,56]. These observations imply that lowered mutation burden due to UV-radiation is indicative of the lower risk of UV-induced skin cancer in African Americans.

In addition to small indels, we also detected large structural variant hotspots in our samples that often coincided with CFSs. For example, rearrangements in FRA7J were found in 14 dif-ferent donors and was the most common hotspot in our samples (S11 Table). Recurrent

breakage at this fragile site has been implicated in the Williams-Beuren syndrome and and this region contains the genes LIMK1, EIF4H(WBSCR1), AUTS2 as well as the tumor suppressor gene FZD9 [57,58]. One explanation for the large number of rearrangements found at a single fragile site is that the genes within this fragile site are preferentially expressed in fibroblasts that may cause transcription-replication collisions often leading to breakage and rearrangements. Alternatively, the replication timing within the fragile locus may be delayed leading to unfinished replication and fragility. Tissue-specific expression and alteration in replication timing at fragile sites has been observed previously in cultured cells [59–61]. As such, we surmise that fibroblast-specific replication-associated difficulties at common fragile sites lead to the formation of rearrangement hotspots in normal skin.

Overall, our work provides an accurate and comprehensive catalog of the somatic genome changes attributable to different DNA damaging processes that act upon human skin cells over the lifetime of the individuals. Our analysis uniquely identifies and measures the impacts of endogenously operating DNA damage, DNA replication errors as well as environmental DNA damage on the somatic mutation load and profiles in each single cell-derived lineage. Finally, we provide the reference for the burden, types and etiologies underlying somatic genome instability in cells of healthy human skin which is required for defining disease level of somatic genome instability.

## Materials and methods

### Ethics statement

Written consent was obtained from all participants in the Environmental Polymorphisms Registry (registered with ClinicalTrials.gov, NCT00341237, and approved by the NIH Institutional Review Board, protocol 04-E-0053). Each participant provided their age, sex and self-identified race.

### Sample collection and processing

4 mm punch skin biopsies were collected from donors' hips. Samples were collected from healthy cancer-free skin. After overnight incubation of the biopsies at 4˚C in 2.66 units/ml dispase (Roche) and 50μg/ml gentamycin (Sigma Aldrich), the epidermis and dermis were separated. The epidermis was emulsified and plated in a six-well cell culture dish in the DermaLife Ma Melanocyte Medium Complete Kit (Lifeline Cell Technology) supplemented with 100μg/ml primocin (Invitrogen). Melanocytes were identified based on their dendritic shape and ability to grow adhered to the dish in serum-free media. The dermis from each biopsy was divided into six to eight pieces which were then allowed to adhere to a six-well cell culture dish and were grown in Dulbecco's modified eagle's medium (Gibco) supplemented with 1X non-essential amino acids (Hyclone), 10% Cosmic Calf Serum (Hyclone), 10% AmnioMax C-100 supplement (Gibco) and 100μg/ml primocin. Fibroblasts were identified as adherent cells elongated in shape that grew from the dermis pieces. All cultures were incubated at 37˚C in a 5% carbon dioxide containing incubator. A portion of bulk cultures of both fibroblasts and melanocytes were harvested for genomic DNA, and another portion was diluted and plated to obtain single cell-derived clones. Fibroblast clones were expanded in culture for 5 to 6 additional passages (4–6 weeks) to obtain ~$10^6$ cells, and genomic DNA was extracted. Genomic DNA extraction from all samples was performed with DNeasy Blood and Tissue kit (Qiagen). Melanocyte clones were expanded in culture for 2 to 3 passages to obtain 10,000 cells and genomic DNA was extracted. 1 to 2.5 ng of the melanocyte genomic DNA was treated with USER (NEB) to remove deaminated cytosines from genomic DNA that are an artifact of DNA extraction [62]. The DNA was amplified using the REPLI-g Mini Kit (Qiagen). 12 to 14

different primer sets were used for PCR across random loci at different chromosomal positions, ranging from 100bp to 500bp, from the amplified genomic DNA. Samples with 10 or more reactions with the correct amplification product were subsequently purified and used for whole-genome sequencing. This quality check allows us to only sequence the genomic DNA with uniform amplification. Venous blood was collected in three to five 8.5mL PAXgene blood DNA tubes (PreAnalytiX/Qiagen) and DNA was isolated from whole blood samples. For 38 DNA samples DNA libraries were prepared using Truseq DNA PCR-free 350bp insert kit (Illumina), and were subsequently sequenced using Illumina HiseqX. For the 30 remaining samples, libraries were prepared using the Nextera DNA Flex library Prep kit (Illumina) and sequenced using the NovaSeq 6000 system. All samples were sequenced as 150 base-paired reads to a depth of 50X to 132X. For a subset of donors, additional skin biopsies were obtained for establishing a second clonal fibroblast lineage.

We also analyzed whole-genome sequenced clonal hip fibroblasts (D1-L-H, D1-R-H1, D1-R-H2, D2-L-H and D2-R-H) from 2 donors that were obtained in a previous study [8].

## Calling somatic genome changes in sequenced clones

The FASTQ reads for each clone and blood sample were aligned to the hg19 genome using the GATK best practices pipeline [63]. Three base substitution callers, SomaticSniper [64], VarScan2 [65,66] and Mutect2 [67] were used to identify the clone-specific mutation calls that were not present in blood of the same donors. Only base substitutions detected by all three callers were analyzed further. Any somatic mutations that were also present in the dbSNP138 database, or any SNVs that overlapped SimpleRepeats tract in the UCSC Genome Browser were removed. The final mutation calls were filtered based on allele frequencies, such that only heterozygous mutations with allele frequencies between 45% and 55% or homozygous mutations with allele frequencies greater than 90% were kept. This methodology of using three independent mutation callers and stringent filtering criteria were used previously for accurate measurements of somatic mutations in human fibroblasts and has demonstrated very high accuracy by orthogonal validations [8]. For bulk whole exome sequencing, Mutect2 was used to call mutations. SNVs that overlapped SimpleRepeats tract or were in the dbSNP138 database were removed. The mutation calls for both whole exome and whole-genome sequencing have been organized as MAF files in the TCGA format and have been submitted to dbGAP study phs001182.v2.p1. Somatic structural variants within 1Mb of each other in different donors were marked as being within a "hotspot".

Delly was used to identify structural variants in the form of deletions, duplications, inversions and translocations [8,68]. Calls which were designated "LowQual" and/or "IMPRECISE" were removed. Clonality of structural variants was determined based on the allelic fraction of reads supporting the variants in the clone. Structural variants with 30% or more reads supporting the structural variant and the absence of any reads supporting the variants in blood were denoted as clonal somatic changes. Due to the low number of variants, we cannot rule out that some of the structural variants may have been generated due to a rearrangement in during the first few cell divisions of the founder cell in culture. However, we think this is unlikely as these cells were passaged less than 3 times before the generation of a clonal lineage. Moreover, the number and types of variants are similar to those detected in previous studies [69–71] indicating that the structural variants detected in our work were likely present in founder cells.

Indels were detected using the tool SV-ABA [72] and were filtered based on multiple criterion. Indel calls with a quality score less than 50 were removed and were only included if they occurred between 45%-55% allele frequency (heterozygous indel), or between 90%-100% allele frequency (homozygous indel). Indels that overlapped with the SimpleRepeats or the RepeatMasker tracts

were removed as these calls were often found to be erroneous. A subset of the indels were visually verified by inspection of the alignments using the Integrative Genomics Viewer [73]. 46 indels were orthogonally verified via PCR amplification and Sanger Sequencing.

## Analyzing base substitution and indel signatures in clones

We used the SigProfilerMatrixGenerator [74] to identify the different types of indels in our samples. SigProfilerExtractor [13] was used to deconvolute the single base substitution, dinucleotide base substitution and indel signatures in our samples. 9 processes with 10 iterations were used within SigProfilerExtractor for extraction of indel and base substitution signatures. MutationalPatterns [75] was used to both identify the cosine similarities between the mutation patterns in samples in this study and the signatures identified in COSMIC and to also identify the contributions of COSMIC signatures in our samples. For base substitutions, the function fit_to_signatures() within MutationalPatterns was used to identify the contributions of the COSMIC signatures on the mutation profile of each sample. For indels, the function fit_to_-signatures() was modified to allow the matrix to have 83 rows instead of 96 so that the indels in our samples could be compared to the known ID signatures (83 channels) in COSMIC.

The enrichment of mutation signatures in each of the samples analyzed in this study was calculated as described in [8,14,35]. For this calculation, context is defined as the +/- 20 bases surrounding the mutated base. The mutated residue is capitalized in the annotation of the signature and the equation to calculate enrichment of a given mutation signature is provided below with the UV-mutation signature yCn➡yTn as an example.

$$Enrichment \ (yCn \rightarrow yTn) = \frac{[Mutations_{yCn \rightarrow yTn}] \times [Context_c]}{[Mutations_{C \rightarrow T}] \times [Context_{ycn}]}$$

For each motif, the reverse complement was also taken into account in the calculations. Mutations <10 bases apart, are excluded in this calculation as these are "complex" mutations that likely arise due to the activity of translesion polymerases and may confound the analysis of the mutation signatures. To determine if increased fold enrichments for the mutation signatures were statistically relevant a Fisher's Exact test was performed wherein the ratio of the number of mutations within the trinucleotide motif ($Mutations_{yCn \rightarrow yTn}$), and those that do not conform to the trinucleotide motif ($Mutations_{C \rightarrow T}$), were compared to the number of unmutated bases in the context that either were in the trinucleotide motif ($Context_{ycn}$) versus those that were not in the context ($Context_c$). Multiple hypothesis testing was further accounted for by the correction of the P-values via the Benjamini-Hochberg method. For samples where enrichment > 1 and the corrected P-value < 0.05, the Minimum Mutation load was calculated for the enriched signature. The equation for calculating this is provided below for the yCn➡yTn mutation signature.

$$Minimum \ Mut \ Load \ (yCn \rightarrow yTn) = \frac{[Mutations_{yCn \rightarrow yTn}] \times [Enrichment_{yCn \rightarrow yTn} - 1]}{[Enrichment_{yCn \rightarrow yTn}]}$$

## Analysis of bulk DNA samples

We also sequenced the exomes of 15 fibroblast bulk samples and 4 melanocyte bulk samples directly cultured from the biopsies. Bulk samples are defined as cells that were not propagated clonally. Libraries were prepared using the Nextera Flex for Enrichment library prep kit, Illumina Exome Panel (Illumina) and IDT for Illumina UD Indexes (Illumina). All samples were sequenced using the NovaSeq 6000 system up to approximately ~150X depth. Somatic mutations were called in the samples by using Mutect2. Whole-genome-sequenced blood samples

from the donor corresponding to each bulk sample was used as a proxy for germline mutations. Only single nucleotide variants called by Mutect2 were further analyzed. Any mutations that were within the dbSNP138 database or in the SimpleRepeats tract were removed.

## Annotation of SNVs

We used Annovar [76] to annotate SNVs for changes to protein sequence using the refGene track from UCSC Genome Browser. Nonsynonymous SNVs or SNVs affecting start or stop codons and splice sites were further annotated using the Cancer Genome Interpreter [39] as driver mutations or passenger mutations.

## Supporting information

**S1 Fig. The distribution of the allele frequencies of the whole genome sequenced clones in this study.** The plots for melanocyte clones are in red. The source data for this figure is in S2 Table.
(TIF)

**S2 Fig.** (A)The mutation spectra in each sequenced clone in this study. The melanocyte clones are marked with an "M" in the X-axis. The source data for this figure are in S2 Table. (B) The NMF-derived mutation signature loads as determined by SigProfilerExtractor in each clone sequenced in this study. Samples form African American donors are annotated with an "A" and melanocyte clones are marked with an "M" in the X-axis. The source data for this figure are in S3 Table.
(TIF)

**S3 Fig. The NMF-derived single base substitution and double base substitution signatures identified in human skin cells.** Total mutations corresponding to the signature in the cohort as determined by SigProfilerExtractor are shown.
(TIF)

**S4 Fig. The correlation of the different UV-specific mutation signatures in the samples in this study.** The total nTt➡nCt mutations and the CC➡TT mutations are plotted against the yCn➡yTn total mutation load in the samples. The black inclined line denotes the linear regression of the data, and the dotted black lines denote the 95% confidence intervals. The source data for this figure are in S4 Table.
(TIF)

**S5 Fig. Comparison of the NMF-derived mutation signatures and the trinucleotide-specific mutation signatures in this study.** The nCg➡nTg minimum mutation load in each sample is plotted against SBS1-associated mutations as determined by SigProfilerExtractor, and the yCn➡yTn minimum mutation load in each sample is plotted against SBS7b-associated mutations. The linear regression of the data is shown, and the dotted lines denote the 95% confidence intervals. The source data for this figure are in S3 and S4 Tables.
(TIF)

**S6 Fig. The distribution of the allele frequencies of the whole exome sequenced bulk samples in this study.** The source data for this figure is in S5 Table.
(TIF)

**S7 Fig. The distribution of the allele frequencies of consensus alleles and cancer drivers.** (A) The allele frequencies of the consensus SNVs identified in the bulk and the corresponding clones are shown. (B) The allele frequency distribution of the cancer driver mutations

identified in the exome of the bulk samples. The source data for this figure is in S5 Table.
(TIF)

**S8 Fig. The NMF-derived indel mutation signatures in this study.** Total mutations corresponding to the signature as determined by SigProfilerExtractor are shown.
(TIF)

**S9 Fig. The analyses of the impact of sex on mutation and indel load in the samples.** The total base substitutions and the total indels in the clonal lineages derived from males and females in this study are shown. A Mann-Whitney U-test was used to determine if the distribution of mutation and indel load were statistically different between the two cohorts. The P-values for the base substitutions was 0.4041, while the P-value for the indels was 0.9401. The source data for this figure is in S10 Table.
(TIF)

**S1 Table. Coverage statistics and donor characteristics for all samples sequenced in this study.**
(XLSX)

**S2 Table. The somatic base substitutions in the whole genome sequenced single skin cell clonal lineages.** a) The fraction of all SNVs prior to filtering that correspond to each allele frequency bin. SNVs that corresponded to allele frequencies between 45% and 55% or above 90% were considered clonal. b) The exonic somatic base substitutions in the samples. c) The mutation spectra in the samples. The reverse complements are considered in the mutation spectra analyses.
(XLSX)

**S3 Table. Agnostic base substitution signature analyses.** a) The contributions of previously determined mutation signatures using MutationalPatterns. b) The number of mutations corresponding to each signature identified by SigProfilerExtractor
(XLSX)

**S4 Table. Motif-specific mutation signature analyses.** a) nCg➜nTg mutation signature analysis b)yCn➜yTn mutation signature analysis. c) nTt➜nCt mutation signature analysis. d) CC➜TT total dinucleotide substitutions.
(XLSX)

**S5 Table. The somatic mutation list for whole exome sequenced bulk tissues.**
(XLSX)

**S6 Table. Somatic indels identified in the samples**
(XLSX)

**S7 Table. The distribution of the types of indels in each sample as identified by SigProfiler-MatrixGenerator.**
(XLSX)

**S8 Table. Agnostic indel signature analysis.** a) The contributions of previously determined mutation signatures using MutationalPatterns. b) The number of mutations corresponding to each signature identified by SigProfilerExtractor
(XLSX)

**S9 Table. The somatic templated insertions identified in human skin fibroblasts and melanocytes.**
(XLSX)

**S10 Table. The somatic genome changes in the samples along with the cell type, and the sex and race of the donors.** a) The total number of somatic genome changes in the samples along with the cell type, and the sex and race of the donors. b) The median values and P-values for Mann-Whitney tests for pairwise comparisons of the somatic genome changes between the sets of clones obtained from White and African American donors.
(XLSX)

**S11 Table. The somatic structural variants identified in the donors.** a) All the somatic structural variants annotated for hotspots, common fragile sites and microhomologies. b) A Fisher's exact test for the use of microhomologies in the SVs that are present within or out of common fragile sites.
(XLSX)

## Acknowledgments

We are thankful to Drs Natasha Degtyareva, Kathleen Hudson, Scott Lujan and Sriram Vijayraghavan for critically reading this manuscript and providing their feedback.

## Author Contributions

**Conceptualization:** Natalie Saini, Dmitry A. Gordenin.

**Data curation:** Natalie Saini, Leszek J. Klimczak, Brian N. Papas, Adam B. Burkholder, Jian-Liang Li, David C. Fargo.

**Formal analysis:** Natalie Saini, Camille K. Giacobone, Leszek J. Klimczak, Dmitry A. Gordenin.

**Funding acquisition:** Dmitry A. Gordenin.

**Investigation:** Natalie Saini, Camille K. Giacobone, Re Bai, Kevin Gerrish, Cynthia L. Innes, Shepherd H. Schurman.

**Methodology:** Natalie Saini, Shepherd H. Schurman, Dmitry A. Gordenin.

**Project administration:** Jian-Liang Li, David C. Fargo, Shepherd H. Schurman, Dmitry A. Gordenin.

**Resources:** David C. Fargo, Kevin Gerrish, Dmitry A. Gordenin.

**Software:** Leszek J. Klimczak.

**Supervision:** Natalie Saini, Dmitry A. Gordenin.

**Validation:** Natalie Saini, Re Bai, Kevin Gerrish, Cynthia L. Innes, Shepherd H. Schurman.

**Visualization:** Natalie Saini, Dmitry A. Gordenin.

**Writing – original draft:** Natalie Saini, Dmitry A. Gordenin.

**Writing – review & editing:** Natalie Saini, Kevin Gerrish, Shepherd H. Schurman, Dmitry A. Gordenin.

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
