## [Decision Letter · Decision Letter 0]

30 Sep 2020

Dear Dr Gordenin,

Thank you very much for submitting your Research Article entitled 'UV-exposure, endogenous DNA damage and DNA replication errors shape the spectra of genome changes in human skin' to PLOS Genetics. Your manuscript was fully evaluated at the editorial level and by three independent peer reviewers. The reviewers appreciated the attention to an important topic and thought that your analysis of UV-induced mutations between different ethnic groups adds novelty to the study. However, they also identified some aspects of the manuscript that should be improved to increase its impact.

Specifically, they asked for additional analyses of some of the data and modifications or additions to several figures. Two reviewers would like to see justifications for the choice of thresholds for allele frequencies. In addition, all felt strongly that the code used during the analysis should be made available on a public repository such as GitHub. Finally, I agree with reviewer 3 that it would be helpful if you could speculate as to the basis behind the surprising finding that UV-induced mutation load is age-independent.

We therefore ask you to modify the manuscript according to the review recommendations before we can consider your manuscript for acceptance. Your revisions should address the specific points made by each reviewer.

In addition, we ask that you:

[LINK]

Sincerely,

Mitch McVey

Guest Editor

PLOS Genetics

David Kwiatkowski

Section Editor: Cancer Genetics

PLOS Genetics

Reviewer's Responses to Questions

**Comments to the Authors:**

Reviewer #1: In the manuscript the authors describe their study of somatic mutations in the normal skin of multiple individuals. For this they rely on cell cloning, i.e., single cells are cultured until colonies are of sufficient size to be sequenced without (or with little) DNA amplification. There were a few previous and similar studies conducted and cited in this manuscript. The novel aspects of this study are: 1) cloning and studying mutations in melanocytes; 2) using samples from different races; 3) using samples from different sexes; 3) a wide age range of participants donated skin samples; 4) analysis that spans beyond just SNVs and includes indels and SVs.

The study reports several significant, new, and interesting findings, of which for me the most interesting were:

* support of UV-induced mutations from by identified indels;

* the experimental evidence that dark skin has protective effect against UV-induced mutations.

I only have a few suggestions on how to improve the manuscript, as well as some minor questions/concerns.

* The threshold of allele frequency for somatic variants in clones “within 45% and 55%” seems quite stringent. Could the authors estimate their sensitivity for discovering the variants from founder cells in a clone?

* Can the authors estimate an average increase in non-UV-induced mutations with age?

* Figure 3B. It is common to show the distribution of indels using negative values for the length of deletions. Please consider this.

* It was unclear which variants were analyzed for cancer drivers. Were there any mutations discovered in clones and in bulks that were predicted to be a driver?

* Can the authors comment about breakpoint features of SVs in CFS and out of CFS? Were there any differences?

* Could the authors comment on the possibility of SVs in the CFS being the results of culturing fibroblasts before deriving clones? I wonder whether CFSs mentioned in the text may have been determined as such from cultured fibroblasts?

* Line 346: Should there be a letter after the first stated CFS “FRA7”?

Reviewer #2: In this work Saini et al extend their previous study (Saini et al, PloS Genetics 2016) on the whole genome sequencing of clonally expanded skin fibroblasts sampled from anatomical sites presumed to have experienced a range of UV exposure. The previous work considered six sampling sites from one donor and four sampling sites from a second donor, both donors of primarily European ancestry. This new work re-analyses the hip-derived samples from the existing data but substantially adds to it expanding the number of donors to 21, including ethnic diversity as a sampling metric and including a limited comparison of mutation patterns between skin fibroblasts and melanocytes.

Analysis includes the quantification of single base substitution signatures, double base changes, indels and structural changes. Several of the results are confirmatory, but importantly so, for example the UV signature mutation load not correlating with patient age and the consistent detection of substantial UV signatures even in sites that are mostly shielded from the sun. However, comparisons of mutation spectra and loads for non-cancerous samples, between ethnicities is to my knowledge novel at the whole-genome multiple sample scale. This reveals a striking reduction in UV signature mutation load for African American samples compared to European – not an unexpected result but important validation of expectation. The authors also find that >4bp deletions also show a relative enrichment in European ancestry skin, suggesting a UV contribution to these multi-basepair deletions. The other key finding is the high frequency of chromosomal rearrangements with breakpoints localised approximately to common fragile sites – again building on earlier observations of the same group but much more solid in this expanded study.

Overall the manuscript is well thought out and presented. The results do provide new insights and the primary data generated is likely to be of interest to multiple groups working on the mechanisms of mutagenesis and those studying skin cancer. To the extent practically possible with human sample data, the experiments are generally well controlled and conclusions justified by the evidence presented. Probably the most important aspect of this work is the comparison of mutation spectra and loads for skin from patients with dramatically different levels of naturally occurring melanin, which is expected to protect from UV damage. A fundamental limitation to the interpretation of these results is that UV exposure per donor is unmeasured and not controlled for. Being hip derived samples the assumption is that these are sites predominantly shielded from the sun, but as such, small differences in exposure (e.g. one bad sunburn) could strongly distort the total lifetime exposure to UV. I think that a caveat/qualification to this effect should be more clearly stated in the discussion.

While I am supportive of publication I suggest the following points could be addressed where practical.

#1. Currently each donor is classified as “White” or “Black or African American”. Within both of these groups there is likely to be a range of skin pigmentation in non-sun-exposed sites (hips). Were measures of that baseline pigmentation taken? It would be useful to report those measures and correlate mutation load/spectra rather than the binary classification of ancestry.

#2. Figure 4 plots include melanocyte clones (the “White” outlier with ~14k SBS mutations must be DAG_H275) but these aren’t discriminated in these plots as they are in figures 1 to 3. They should be indicated, particularly when they represent outlier data-points. Do the significant differences noted in these plots remain significant after excluding the melanocyte clones?

#3. A useful complement to the existing plots would be a plot summarising: (a) SBS mutation load per clone, (b) contribution of mutation signature contribution per clone like Fig S1 but with a richer set of signatures such as COSMIC, (c) annotation of ethnicity and cell type (fibroblast/melanocyte) per clone. Such a plot, possibly rank-ordered by total mutation load, would be an informative summary of the key data generated and would make apparent some results that appear to be masked in the current presentation. For example the clone DAG_H275 has a high SBS load and from Fig S1 looks to be dominated by C→T substitutions consistent with UV exposure, but then comparison to Fig 4a (highest purple value to highest red value) and mutation counts in table S2 indicate that this clone must dominated by rCn to rTn mutations (COSMIC signature SBS2). Is this an outlier amongst melanocyte clones?

#4. Figure 3b. It is difficult to see the individual histogram bars, especially at the left edge. Suggest that this plot would be more informative if points were used rather than histograms and the x-axis shown on a log scale (in addition to the y-axis being kept on a log scale).

#5. Figure S2. X-axis annotations are not clear (partially overlapped by subsequent panels).

#6. Code availability: “The R-code for analysis of the trinucleotide-specific mutation signatures is the same as used in [8] and will be provided on request.” For efficient and long-term reproducibility code, especially that being used across multiple publications, it should be in a pubic repository such as GitHub, Figshare, Zendo or an institutional repository that provides a DOI.

#7. Data availability: In the additional information front section of the submission, the authors state “Yes - all data are fully available without restriction”. This should also be clearly stated in the manuscript. I would also suggest that the work will be more widely used and cited if the full list of somatic mutation calls (e.g. MAF files filtered for germline variants) can be made available without the requirement for DAC approval, which although an important process for limiting access to potentially personally identifiable germline genotypes, is unnecessary limiting for somatic mutations.

Reviewer #3: The manuscript by Saini et al. provides a very interesting analysis of mutation signatures in primary skin cells in donors of different ages and races. As such, it is of great interest to the scientific community. I have several major and minor points for authors to address.

Major points:

How did sequencing errors affect the results?

How did whole-genome amplification, and potential errors and biases, affect the results?

Please justify the choice of 45%-55% and 90% threshold for allele frequencies used. Were simulations used to derive these?

The finding that UV-induced mutation load is independent of age is counter-intuitive. Please provide an explanation behind this unexpected result.

Also, how do the authors explain that UV-based substitution signatures were prominent in many samples obtained from sun-shielded skin?

The paper should be read by a native speaker of English to fix some minor grammar issues.

The authors should provide all their code on GitHub.

The authors should provide a file with all the genetic variants they identified including their genomic locations for all the samples.

Can the authors speculate about the applications of their study with respect to the increased risk of skin cancer in individuals with high UV exposure?

Minor points:

The title should read 'UV exposure, endogenous DNA damage, and DNA replication errors shape the spectra of genome changes in human skin'

Abstract: usually ‘mutation load’ is used as singular

Abstract: ‘…DNA replication stalling at common fragile sites Is a potent source…'

Abstract, last sentence: please change ‘intrinsic factors’ -> ‘endogenous factors’

P. 3, line 39: ‘Large scale’ -> ‘Large-scale’

P. 3, line 44: there should be no comma after ‘Since’

P. 4, line 57: add space after ‘Pol’ and before the greek letter of the polymerase

P. 4, line 59: ‘Error-prone’

P. 4, line 69: ‘UV radiation’

P. 4, line 71: add comma after ‘melanocytes’

P. 4, line 76: ‘burdens’ -> ‘burden’; ‘sun shielded’ -> ‘sun-shielded’

P. 5, line 82: add comma after ‘skin’, change ‘signature’ -> ‘mutation signature’

P. 5, line 86: add comma after ‘(InDels)’

P. 5, line 89: change to ‘…due to either small sample sizes or difficulties…’

P. 6, line 101: add comma after ‘errors’

P. 6, line 114: remove ‘-‘ after ‘cell’

P. 6, line 116: here and elsewhere in the paper spell out all numbers below 10

P. 6, lines 118 and 121: ‘whole-genome’

P. 7, line 122: ‘whole-genome’

Figures 1-3, 5: ‘R-square’ -> ‘R-squared’

P. 7, line 135: ‘sequencing coverage’ -> ‘sequencing depth’ (was this average per site?)

P. 7, line 137: ‘variations’ -> ‘variants’

P. 8, line 163: ‘see’ -> ‘saw’

P. 9, line 169: add comma after ‘SBS5’

P. 9, line 173: ‘that’ -> ‘this’

P. 9, line 179: ‘SBS1-associated’

P. 10, line 189: please add a reference after ‘have previously been shown to be associated with UV-induced DNA damage in human cells’

P. 10, line 206: ‘Whole-exome’

P. 12, line 240: remove comma after ‘Since’

P. 12, line 247: ‘spanning 5 nucleotides or larger’ -> ‘spanning five or more nucleotides’

P. 12, line 248: ‘have microhomology of one or more bases’

P. 12, line 252: ‘deletions of 5 bases or more’ -> ‘deletions of five or more bases’

P. 13, line 254: ‘spanning five or more nucleotides’

P. 13, line 256: ‘indel load'

P. 13, line 257, and elsewhere in the manuscript: ‘InDel’ -> ‘indel’

P. 14, line 290 and elsewhere in the manuscript: ‘gender’ -> ‘sex’

P. 14, line 297: ‘do see’ -> ‘did see’

P. 16, line 324: ‘whole-genome’

P. 16, lines 327 and 338: add space before ‘bp’ and ‘Mbp’

P. 17, line 351: ‘surmise -> ‘hypothesize’

P. 18, line 356: ‘Whole-genome’

P. 18, line 373: ‘endogenously operating’ -> ‘endogenous’

P. 20, line 416: ‘loads’ -> ‘load’

**Have all data underlying the figures and results presented in the manuscript been provided?**

Reviewer #1: Yes

Reviewer #2: Yes

Reviewer #3: **No: **The authors should provide all their code on GitHub.

The authors should provide a file with all the genetic variants they identified including their genomic locations for all the samples.

PLOS authors have the option to publish the peer review history of their article (what does this mean?). If published, this will include your full peer review and any attached files.

Reviewer #1: No

Reviewer #2: **Yes: **Martin S Taylor

Reviewer #3: No

---

## [Decision Letter · Decision Letter 1]

7 Dec 2020

Dear Dr Gordenin,

We are pleased to inform you that your manuscript entitled "UV-exposure, endogenous DNA damage, and DNA replication errors shape the spectra of genome changes in human skin" has been editorially accepted for publication in PLOS Genetics. Congratulations! The reviewers all agreed that your revisions address their concerns and that the manuscript will make an important contribution to the somatic mutation field. As the managing editor, I agree with this sentiment. Please note that Reviewer 1 did request one additional wording change when you submit the final version.

Yours sincerely,

Mitch McVey

Guest Editor

PLOS Genetics

David Kwiatkowski

Section Editor: Cancer Genetics

PLOS Genetics

Comments from the reviewers (if applicable):

Reviewer's Responses to Questions

**Comments to the Authors:**

Reviewer #1: The authors did adequately respond to my comments. I would just like to comment on the following statement: “We did not see such a peak in the whole-genome amplified melanocyte clones which could reflect uneven genome amplification and localized genome duplications during the whole-genome amplification step.” The in vitro amplification step is likely to make distribution of allele frequencies broader, but the distribution still should be centered at 50%. Based on Figure S1, a possible interpretation is that some of melanocyte colonies have two founder cells and thus the corresponding distributions have peaks at 25% frequency. In this regard, I’m glad that the authors are careful in describing their results with the statement “… allows us to estimate the minimum accurate number of somatic mutations in the founder cells,” but, in light of possible technical challenges, I would remove word “accurate” from the statement.

Reviewer #2: I am happy that my previously raised points have been addressed to the extent practically possible. To my reading the points raised by other viewers also seem to have been well addressed by the revisions to the manuscript. I'm happy to recommend publication, it's a good well thought out study that complements a recent flurry of papers on somatic mutation in normal (non-cancer) tissue. Clearly a hot area in which this paper still has it's unique selling points including the stratification by ancestry/pigmentation.

Reviewer #3: The authors have adequately addressed my concerns.

**Have all data underlying the figures and results presented in the manuscript been provided?**

Reviewer #1: Yes

Reviewer #2: Yes

Reviewer #3: None

PLOS authors have the option to publish the peer review history of their article (what does this mean?). If published, this will include your full peer review and any attached files.

Reviewer #1: No

Reviewer #2: **Yes: **Martin S. Taylor

Reviewer #3: No

**Data Deposition**

http://datadryad.org/submit?journalID=pgenetics&manu=PGENETICS-D-20-01306R1

**Press Queries**

---

## [Editor Report · Acceptance letter]

23 Dec 2020

PGENETICS-D-20-01306R1 

UV-exposure, endogenous DNA damage, and DNA replication errors shape the spectra of genome changes in human skin 

Dear Dr Gordenin, 

We are pleased to inform you that your manuscript entitled "UV-exposure, endogenous DNA damage, and DNA replication errors shape the spectra of genome changes in human skin" has been formally accepted for publication in PLOS Genetics! Your manuscript is now with our production department and you will be notified of the publication date in due course.

With kind regards,

Melanie Wincott

PLOS Genetics

On behalf of:
